# Pharmacoeconomic Profiles of Advanced Therapy Medicinal Products in Rare Diseases: A Systematic Review

**DOI:** 10.3390/healthcare13151894

**Published:** 2025-08-02

**Authors:** Marianna Serino, Milana Krstin, Sara Mucherino, Enrica Menditto, Valentina Orlando

**Affiliations:** 1Department of Pharmacy, University of Naples Federico II, 80131 Naples, Italy; marianna.serino@unina.it (M.S.); milana.krstin@unina.it (M.K.); enrica.menditto@unina.it (E.M.); valentina.orlando@unina.it (V.O.); 2Center of Pharmacoeconomics and Drug Utilization Research (CIRFF), University of Naples Federico II, 80131 Naples, Italy

**Keywords:** Advanced Therapy Medicinal Products, ATMP, gene therapy, rare diseases, cost-effectiveness, economic evaluation

## Abstract

**Background and aim**: Advanced Therapy Medicinal Products (ATMPs) are innovative drugs based on genes, tissues, or cells that target rare and severe diseases. ATMPs have shown promising clinical outcomes but are associated with high costs, raising questions about cost-effectiveness. Hence, this systematic review aims to analyze the cost-effectiveness and cost-utility profiles of the European Medicines Agency-authorized ATMPs for treating rare diseases. **Methods**: A systematic review was conducted following PRISMA guidelines. Studies were identified by searching PubMed, Embase, Web of Science, and ProQuest scientific databases. Economic evaluations reporting incremental cost-effectiveness/utility ratios (ICERs/ICURs) for ATMPs were included. Costs were standardized to 2023 Euros, and a cost-effectiveness plane was constructed to evaluate the results against willingness-to-pay (WTP) thresholds of EUR 50,000, EUR 100,000, and EUR 150,000 per QALY, as part of a sensitivity analysis. **Results**: A total of 61 studies met the inclusion criteria. ATMPs for rare blood diseases, such as tisagenlecleucel and axicabtagene ciloleucel, were found to be cost-effective in a majority of studies, with incremental QALYs ranging from 1.5 to 10 per patient over lifetime horizon. Tisagenlecleucel demonstrated a positive cost-effectiveness profile in the treatment of acute lymphoblastic leukemia (58%), while axicabtagene ciloleucel showed a positive profile in the treatment of diffuse large B-cell lymphoma (85%). Onasemnogene abeparvovec for spinal muscular atrophy (SMA) showed uncertain cost-effectiveness results, and voretigene neparvovec for retinal diseases was not cost-effective in 40% of studies, with incremental QALYs around 1.3 and high costs exceeding the WTP threshold set. **Conclusions**: ATMPs in treating rare diseases show promising economic potential, but cost-effectiveness varies across indications. Policymakers must balance innovation with system sustainability, using refined models and the long-term impact on patient outcomes.

## 1. Introduction

Advanced Therapy Medicinal Products (ATMPs) were developed as innovative biological products that directly target the cause of the disease and significantly improve health outcomes of patients who face neoplasms, metabolic, blood, cardiovascular, neurologic, retinal, and rare diseases [1]. The Committee for Advanced Therapies (CAT), established by the European Medicines Agency (EMA), has classified ATMPs as gene therapies, somatic-cell therapies, tissue-engineered medicines, and combined ATMPs [2]. In 2008, the Advanced Medicines Regulation was implemented, regulating the development and marketing authorization of ATMPs, providing an immense opportunity to treat rare diseases [3]. The definitions and classification according to Regulation (EC) No 1394/2007 and Directive 2001/83/EC are represented in Appendix A. The centralized authorization process is carried out through the CAT, and is responsible for the scientific evaluation, classification, safety, efficacy, and quality estimate, and the Committee for Medicinal Products for Human Use (CHMP) provides the final decisions based on submitted opinions [4]. In 2024, 20 ATMPs were already authorized within the European Union market, 16 with orphan designation proposed for rare disease treatment [5]. ATMPs are associated with elevated costs, largely due to their innovative, often one-time administration; especially in the case of ATMPs for ultra-rare diseases, the cost per patient tends to be even higher than for rare diseases. However, it remains unclear whether these prices are justified by clinical outcomes [6]. In this sense, economic evaluations are crucial to ensure that innovative and advanced treatments provide value for money while considering budget constraints [7,8]. However, this can be challenging due to the following methodological issues: limited size and design of trials, uncertain data on long-term effects, uncertain efficacy assessment, limited data on estimating impact on quality of life, and reduced generalizability [7]. Furthermore, to inform healthcare payers, there is a great need for robust economic evidence in ATMP therapies [7]. Despite this, to date, there is a lack in the literature addressing an overview of the real cost-effectiveness profile of ATMPs for rare diseases, considering their sustained therapeutic and clinical outcomes. Hence, using cost-effectiveness principles may assess the health and cost impact of ATMPs, compared to standard treatments, and assist in economic allocation strategies considering budget constraints. Therefore, this study aims to create an overview of the current state of the art regarding the cost-effectiveness and/or cost-utility profiles of ATMPs for rare and ultra-rare diseases.

## 2. Materials and Methods

### 2.1. Protocol, Registration, and Reporting

The protocol for this review was registered with the International Prospective Register of Systematic Reviews (PROSPERO, reference CRD42023472088). A systematic review was conducted, and its reporting followed the Preferred Reporting Items for Systematic Reviews and Meta-Analyses (PRISMA 2020 statement guidelines) [9].

### 2.2. Eligibility Criteria, Information Sources, and Search Strategy

The literature search was conducted on 4 October 2024, according to the eligibility criteria detailed in Table 1. During the literature research, the following databases were used: PubMed (MEDLINE), Embase, Web of Science, and ProQuest. The search strategy included a combination of headings and keywords to delineate the target, Patient, Intervention (check Appendix A), Comparator, Outcomes, and Study design (PICOS), criteria, utilizing MeSH (Medical Subject Headings) or Emtree terms, as appropriate, and searching within the title and abstract fields detailed in Appendix A. We conducted a targeted three-stage search using brand names of ATMPs, their generic names, and pharmacoeconomic terms such as “cost-effectiveness,” “cost–utility,” “ICER,” “ICUR”, “QALY,” and “economic evaluation.” Boolean operators AND/OR were utilized to refine the search parameters. Search strategy and syntax are detailed in the Appendix A.

### 2.3. Study Selection, Data Extraction, and Data Synthesis

The literature search and screening process, which was based on the PICOS question, was independently conducted by two researchers (M.S. and M.K.) using a double-blind process. To assess the level of agreement between reviewers during the study selection phase, the Cohen’s kappa coefficient was calculated. All records were independently evaluated by both reviewers, resulting in a substantial agreement (Cohen’s kappa = 0.78), indicating reliable inter-rater consistency. Discrepancies were resolved through discussion or by involving a third reviewer (S.M.). After the duplicate studies were removed, the titles and abstracts of identified studies were reviewed, selecting those suitable for further assessment. Afterward, the full text of the identified studies was reviewed to ensure the completeness of all criteria (Figure 1). The differences that arose during the research were discussed, and the opinion of the other two researchers (E.M. and V.O.) was sought. The following data were extracted from each record identified: country of the paper, year of publication, INN name of the evaluated ATMP, treatment comparator, therapeutic indication (diagnosis) to which medicine was assessed, incremental cost-effectiveness/cost-utility ratio, as well as whether ATMP cost-effective profile (see Appendix A). Additionally, methodological aspects of all the economic evaluations included were evaluated by extracting data regarding study design, type of costs considered (direct and indirect), costs’ perspectives, willingness-to-pay (WTP) threshold of the country considered in the study, and reference year of costs, as well as applied discount rate, the presence of sensitivity analyses, and time horizon of the analysis (see Appendix A).

### 2.4. Quality Assessment

The quality of papers and assessment was evaluated using a checklist proposed by Olry de Labry-Lima et al. [10]. The checklist was compiled using a questionnaire composed by ISPOR to assess the credibility and relevance of economic modeling studies [11,12]. It also incorporates guidelines for conducting a systematic review of economic evaluation [11,12]. Through the use of the checklist, the adequacy of all studies was assessed, including their methodological soundness and conclusions. This evaluation covered several domains of design (eligible population, intervention, and comparator, and adequate study design, appropriate time horizon, and viewpoint study perspective), data analysis (resources, unit costs, relevant outcome, main measure of effect, and sensitivity analysis), interpretation, generalizability, and conflict of interest [10]. A detailed explanation of each domain is shown in Appendix A. Compared to the CHEERS checklist, the checklist used in this study also includes reporting information of the following: study population, setting and location, comparators, perspective, time horizon, discount rate, selection, measurement and valuation of outcomes, currency, rationale and description of the model, analytics and assumptions, characterizing uncertainty, characterizing heterogeneity, conflict of interest, and generalizability. In addition, each study was assessed for a wide range of potential biases. These included selection bias, study design bias, measurement bias, data availability bias, extrapolation bias, modeling bias, assumption bias, uncertainty bias, data source bias, omission bias, and misclassification bias. Because many of the included studies were sponsored by pharmaceutical companies, a separate column indicating any potential conflicts of interest was included. This was performed in cases where the company that funded the study had at least one employee listed as an author. To assess the robustness of our primary outcome, we excluded studies with a high or unclear risk of bias. Studies scoring above an arbitrary threshold of 75% were considered to have higher reporting quality [7]. The results are represented as moderate/low (the effect is likely to be substantially different from the estimated effect), moderate (moderately confident in the estimated effect), moderate/high (the effect is likely to be close to the estimated effect), or high (effect lies close to that of the estimated effect estimate).

Quality assessment analysis was conducted by two reviewers (M.S. and M.K.) independently, and any discrepancies were discussed.

### 2.5. Statistical Analyses

The incremental cost-effectiveness ratio (ICER) and incremental cost-utility ratio (ICUR) were identified and extracted for each included study to summarize the estimated cost-effectiveness for each ATMP. In studies where the pharmacoeconomic profile is not explicitly stated but expressed in percentages, without explicit information about pharmacoeconomic profile, we considered any profile exceeding 50% to be cost-effective; otherwise, it was classified as negative. Also, for studies that did not explicitly report ICER/ICUR values, these were calculated as the difference in costs between the intervention and the comparator, divided by the difference in their effectiveness, representing the additional cost per additional unit of health benefit (e.g., QALY) gained. All identified ICER/ICUR values were standardized in a single currency (Euro). To ensure comparability, all costs were adjusted for inflation and converted into 2023 Euros using the CCEMG-EPPI cost converter tool (2024 version). CCEMG–EPPI represents a two-stage tool. In the first stage, the original cost is adjusted to 2023 price levels using a gross domestic product deflator index (‘GDPD values’), which is seen as a measure of overall inflation, considering price changes across many different economic sectors. Additionally, in the second stage, the calculator was used to convert the 2023 price–year adjusted cost estimate from the original currency to Euros using conversion rates based on purchasing power parity (PPP) for GDP. This makes currency conversion more accurate, so costs are compared using the same average international standards, which is not based just on exchange rates but rather the actual cost in each country [13]. In cases where the base year was not specified in the study, the year of publication was used.

A qualitative analysis of all ATMPs’ CE profiles identified was represented with a Sankey flow diagram grouping for ATMPs and their approved indications. Finally, given well-documented concerns about a meta-analysis of cost-effectiveness studies (including heterogeneity in resource utilization and pricing across contexts), no meta-analysis was attempted for this study [14,15,16,17].

Cost-effective profiles of ATMPs for rare diseases were compared on a cost-effectiveness plane. Results were plotted in terms of incremental QALYs (*x*-axis) and incremental costs relative to alternative treatments (*y*-axis). To set a common threshold for the cost-effectiveness profile definition, which considers substantial differences in willingness to pay (WTP) across global settings, a sensitivity analysis was carried out using three WTP thresholds: EUR 50,000/QALY, EUR 100,000/QALY, and EUR 150,000/QALY. These high thresholds were also set considering the Highly Specialised Technologies (HST) program, established by the British National Institute for Clinical Excellence (NICE), which recommends higher WTPs for very rare conditions.

All screening and data management tasks were performed using Microsoft Excel, version 2406 (Microsoft Corporation, Redmond, WA, USA), while Sankey diagrams were created with the Sankey MATIC tool [18]. The cost-effectiveness plane was generated using the R programming language for statistical computing (R and R Studio, version 4.4.2).

## 3. Results

### 3.1. The Literature Search and Screening Results

The search across the four databases queried identified 1460 relevant papers. As shown in Figure 1, n = 527 duplicates were removed, and n = 933 titles and abstracts were screened, yielding n = 441 eligible studies. Additionally, n = 337 papers without full-text availability were removed. During the full-text screening stage, we excluded all publication types that did not meet the eligibility criteria. Reports assessed for eligibility were n = 64, and among these, n = 3 were excluded due to a lack of information on ICER/ICUR or QALY. The final number of studies included in the systematic review was n = 61 (Figure 1).

### 3.2. Quality Assessment Results

The assessments of the quality of evidence were conducted using the checklist from Olry de Labry-Lima et al. [10], 2023 (Appendix A). Studies that scored very low quality (n = 3), due to the lack of the necessary information according to the PICOS, were excluded from the analysis (Figure 1). Overall, among the included studies (n = 61), most economic evaluations (54%, n = 33 studies) showed moderate-to-high certainty of evidence and strength of recommendations. Furthermore, 11% (n = 7) of the studies achieved a high certainty rating, while 33% (n = 20) a moderate rating, and 2% (n = 1) a moderate/low. Regarding the latter, lower scores are surely related to the intrinsic nature of clinical trials of orphan drugs for rare disease treatment due to the small sample sizes, the single-arm design methods, and so on. All these factors can potentially impact overall efficacy in cost-effectiveness estimation. Furthermore, the quality of the evidence itself was affected by the lack of information about the comparator and reference year in the cost-effectiveness analysis; these factors can impact the conclusion. Some of the most frequently identified potential risks of bias are assumption bias (39%), study design bias (31%), and extrapolation bias (37%). In 55% of the analyzed studies, a potential conflict of interest was identified due to at least one author being affiliated with the funding company (Appendix A).

### 3.3. Overall Characteristics of Studies Included in the Review

The global geographical distribution of the n = 61 economic evaluation shows a significant majority of studies carried out in the United States (41%; n = 26) (Figure 2). A minority of studies were conducted in European settings, e.g., Spain (5%; n = 3), Switzerland (3%; n = 2), Netherlands (3%; n = 2), Ireland (3%; n = 2), Germany (3%; n = 2), France (3%; n = 2), Italy (2%; n = 1), and Sweden (2%; n = 1) (Figure 2). Additionally, 93% of the included studies use a lifetime time horizon, and some studies simulate the horizon until the remaining surviving patients reach 100 years of age, which is the assumption of lifetime. All identified studies used discount rates that are consistent with the guidelines of the countries in which the studies were conducted. The highest discount rate, up to 5%, was observed in countries such as Australia and China. All methodological information of the overall included studies was extracted and reported in Appendix A.

Overall, the Sankey diagram, shown in Figure 3, illustrates the overall distribution of the pharmacoeconomic profiles of ATMPs and their approved indication(s) detected from the included studies. The width of each flow demonstrates the number of included economic evaluations, and ATMPs are presented in descending order, specifically from axicabtagene ciloleucel (n = 23 studies) [19,20,21,22,23,24,25,26,27,28,29,30,31,32,33,34,35,36,37,38,39,40,41] to etranacogene dezaparvovec (n = 1 study) [42]. The second nodes represent the approved indications of ATMP. Moreover, 85% [19,20,22,23,24,25,26,27,29,31,32,33,34,35,36,37,38] of the studies assessed axicabtagene ciloleucel as cost-effective compared to alternatives in diffuse large B cell lymphoma (DLBCL) treatment, while 66% (n = 2) assessed a positive cost-effective profile in follicular lymphoma (FL) treatment [40,41]. Also, 58% (n = 7) of included studies demonstrated a positive cost-effective profile of tisagenlecleucel in acute lymphoblastic leukemia (ALL) treatments [43,44,45,46,47,48,49], while brexucabtagene autoleucel demonstrated a 100% (n = 5) positive pharmacoeconomic profile in mantle cell lymphoma (MCL) and ALL treatments [50,51,52,53,54], the same percentage showed etranacogene dezaparvovec in the hemophilia B treatment [42]. Furthermore, 75% (n = 3) of included economic evidence concluded a positive pharmacoeconomic profile of ciltacabtagene autoleucel in multiple myeloma (MM) treatment [55,56,57], while 60% (n = 3) of economic evidence demonstrated voretigene neparvovec as a cost-effective option in RPE65-associated inherited retinal disease (RPE65-IRD treatment) [58,59,60].

### 3.4. Cost-Effectiveness Profiles of the Overall Studies

Figure 4 illustrates the results plotted into a cost-effective plane. In more depth, most of the included studies are in the northeast quadrant (positive incremental costs and positive incremental QALY assessing ATMP as the therapeutic choice that gains more health benefits, but also more expenses compared to alternatives) [19,21,22,23,24,25,26,27,28,29,30,31,32,33,34,35,36,38,39,40,41,43,44,45,46,47,48,49,50,51,52,53,54,55,56,57,59,60,61,62,63,64,65,66,67,68,69,70,71,72,73,74,75,76,77,78,79]. Studies falling below the threshold of EUR 100,000/QALY consider ATMPs cost-effective compared to the alternative therapies [19,22,24,25,27,29,31,33,34,35,36,38,40,41,43,44,45,46,47,48,49,50,52,53,54,55,62,63,65,67,69,72,73,78,79]. Specifically, studies evaluating the cost-effectiveness of tisagenlecleucel are located below the WTP of EUR 100,000/QALY, making them a viable therapeutic choice, with incremental QALY often over five, but they also have significantly higher incremental costs [43,44,45,46,48,78]. On the contrary, axicabtagene ciloleucel represents a therapeutic option with lower incremental QALY but also notably lower incremental costs [21,22,23,24,26,27,28,29,30,31,32,33,34,39,40,41]. Additionally, it is noticeable that studies of this ATMP estimate very similar ICERs, leading to a large concentration at one point. The six studies in the southeast quadrant (negative incremental cost and positive incremental QALY) represent the dominant therapeutic choices, with higher effectiveness but also lower costs [20,37,42,58,75,77]. Also, four studies evaluating the cost-effectiveness of voretigene neparvovec in retinal disease treatment represent therapy outside the WTP of EUR 100,000/QALY, with low incremental QALY but high incremental costs [59,60,66,73]. Despite most references focusing on incremental QALY up to two and low incremental costs, some studies are outliers, and these are most commonly examining the cost-effectiveness of onasemnogene abeparvovec [68,76] and only one study that estimates etranacogene dezaparvovec in hemophilia treatment [42]. Unlike etranacogene dezaparvovec, which represents a dominant outlier, onasemnogene abeparvovec has extreme values in both the dominant square and the northeast quadrant.

### 3.5. Cost-Effectiveness Profiles of ATMPs for Rare Blood Cancer and Hemophilia B Treatment

The summary of the results, including the analysis of ATMPs and their therapeutic indication, is provided in this study. Table 2 presents the mean ICER and mean incremental QALY, excluding outliers due to significant variations. Moreover, as shown in Figure 4, the majority of studies examining the pharmacoeconomic profile of axicabtagene ciloleucel are concentrated below the WTP threshold of EUR 100,000/QALY; however, one study is observed in the dominant quadrant. Among n = 20 studies evaluating the cost-effectiveness of axicabtagene ciloleucel in the treatment of DLBCL [19,20,21,22,23,24,25,26,27,28,29,30,31,32,33,34,35,36,37,38], 85% (n = 17) of them demonstrated cost-effectiveness compared to the alternatives [19,20,22,23,24,25,26,27,29,31,32,33,34,35,36,37,38], considering the mentioned threshold. Also, axicabtagene ciloleucel represented a cost-effective strategy as a second-line DLBCL treatment in n = 5 [23,32,33,34,38] evaluated studies, while the other n = 5 studies also proved its cost-effectiveness in the subsequent treatment lines [20,23,30,32,35]. From the total amount of included studies, seven studies were identified that lie below the EUR 50,000/QALY WTP threshold [29,30,31,32,35,36,38]. Two studies even exceed the highest range in the sensitivity analysis of EUR 150,000 per QALY [21,30]. As shown in Figure 4, axicabtagene ciloleucel presented an unfavorable cost-effectiveness profile in n = 3 evaluated studies [28,30,32], of which Wu et al. [30] concluded in the first, second, and third lines of therapy compared with the alternatives, taking into account the threshold of EUR 29,449/QALY. In addition to its established role in treating DLBCL, axicabtagene ciloleucel has shown promise in the potential treatment of FL [39,40,41], where 66% (n = 2) [40,41] of included studies demonstrated cost-effectiveness compared to alternatives (mosunetuzumab, SoC), considering a threshold of EUR 105,750/QALY. Moreover, Oluwole [40] and Potnis [39] have estimated their cost-effective profile in third-line FL treatment, leading to completely different conclusions in the same settings from the same perspective, but by using different types of models (such as partitioned survival versus the Markov model; differences in assumptions on survival extrapolations, specific utility values, and cost inputs, see Appendix A). Also, the median ICER for axicabtagene ciloleucel is EUR 64,540/QALY in DLBCL and EUR 66,829/QALY in FL. As presented in Figure 4, n = 13 [43,44,45,46,47,48,49,65,67,69,72,78,79] studies examining the cost-effectiveness of tisagenlecleucel in treating rare blood cancer diseases are located below the set threshold of EUR 100,000/QALY. Out of the 13 studies mentioned, nine fall within the cost-effectiveness threshold of EUR 50,000/QALY [43,45,46,47,48,49,65,69,79]. Additionally, two studies [23,30] are very close to the quadrant that promotes the dominance of older therapies over the mentioned ATMP. Regarding Appendix A, seven economic evaluation studies of tisagenlecleucel [43,44,45,46,47,48,49] demonstrated an affordable pharmacoeconomic profile in treatment compared to the standard of care, clofarabine, and blinatumomab. As shown in Appendix A, n = 3 did not estimate a cost-effective profile [65,67,72], considering that Carey [63] conducted his research on a pediatric and younger patient population, with an estimated ICER EUR 71,370/QALY on the WTP threshold of EUR 43,943/QALY. Notably, Wang [49] and Choe [23] estimated tisagenlecleucel is likely to be a cost-effective and lifesaving strategy as the third line for ALL therapy with ICERs EUR 47,222/QALY and EUR 53,545/QALY from Singapore (WTP threshold EUR 105,750/QALY) and EUR 99,012/QALY from US the perspective (WTP threshold EUR 117,318/QALY). The assessment of the cost-effectiveness profile of tisagenlecleucel in DLBCL treatment is still uncertain, with an equal number of studies showing that the therapy is cost-effective (n = 5) [23,48,69,70,79] and that it is not [23,30,61,63,71]. The median ICER for tisagenlecleucel in ALL treatment is EUR 49,081/QALY, while in DLCBL treatment the amount is higher, amounting to EUR 68,098/QALY.

Figure 4 also shows that ciltacabtagene autoleucel for the MM treatment seems to be a cost-effective strategy (n = 3) in studies below the WTP threshold of EUR 100,000/QALY [55,56,57]. On the other side, the CE profile of idecabtagene vicleucel for the same indication seems to be uncertain [55,56,57,74]. For MCL therapeutic indication, brexucabtagene autoleucel is also considered a cost-effective option compared to the alternative therapy, with a range of ICERs varying from EUR 26,140/QALY to EUR 76,656/QALY, while ICUR is slightly higher: EUR 67,720/QALY [50,51,52,53,54]. Also, lisocabtagene maraleucel demonstrated an uncertain pharmacoeconomic profile, with a low number of included studies [30,75].

Regarding the hemophilia B treatments, etranacogene dezaparvovec is approved for this indication and represents a cost-effective alternative in German settings with a WTP of EUR 50,000/QALY. ATMP does not lead to a significantly higher incremental QALY, but it results in savings of EUR 1,200,000, representing a dominant treatment (Figure 4) [42].

Axicabtagene ciloleucel demonstrated an affordable pharmacoeconomic profile in first, second, third, and subsequent lines of treatment for DLBCL, as well as in follicular lymphoma. In contrast, tisagenlecleucel, while showing a favorable pharmacoeconomic profile in DLBCL, did not demonstrate strong cost-effectiveness in FL. Meanwhile, brexucabtagene autoleucel demonstrated a favorable pharmacoeconomic profile in MCL, and ciltacabtagene autoleucel showed similar promise in MM, whereas idecabtagene vicleucel did not exhibit a favorable profile for the same indication.

### 3.6. Cost-Effectiveness Profiles of ATMPs for Spinal Muscular Atrophy Type I Treatment

Onasemnogene abeparvovec, representing ATMP approved for serious, rare diseases, such as spinal muscular atrophy type I, demonstrated significant variations in its pharmacoeconomic profile depending on settings (Figure 4). Specifically, n = 2 studies are substantially above the WTP of 100,000/QALY (not cost-effective) [68,76]. In contrast, n = 3 studies are below the threshold of cost-effective, with n = 1 study falling into the dominant quadrant (ICER EUR −53,073/QALY) [77] (see Figure 4). Two studies were identified that fall below the set threshold of EUR 50,000/QALY, both studies identify a similar incremental QALY of approximately 11, while showing a significant difference in incremental cost. They were conducted within the same healthcare setting and used nusinersen as the comparator. The study conducted by Wang et al. is the only one that exceeds the highest defined threshold of EUR 150,000 per QALY, considering both comparators, but the application of the highest discount rate for both cost and effectiveness of as much as 5% may be the reason, which is in line with Australian HTA guidelines [76]. Median ICER of onasemnogene abeparvovec amounts to EUR 94,492/QALY.

Also, a variation in the profiles is observed depending on the comparator. Most studies in which the comparator was nusinersen demonstrate a positive cost-effectiveness profile [62,68,77], while in all studies with BSC or SoC as comparators the profile is negative [62,68,76,77]. The same ATMP, onasemnogene aberparovec, when compared to nusinersen, reported a wide ICER ranging from EUR 25,979/QALY to EUR 55,152/QALY [62,68,76].

In contrast to all other analyzed ATMPs, onasemnogene abeparvovec demonstrated the highest sensitivity to the choice of comparator, discount rate, and time horizon.

### 3.7. Cost-Effectiveness Profiles of ATMPs for Retinal Diseases with RP E65-Mutation Treatment

Voretigene neparvovec, a novel therapy approved for IRD patients with a rare biallelic RPE65 mutation, was cost-effective compared to the standard of care in n = 3 studies [58,59,60] with a WTP threshold above EUR 100,000/QALY (i.e., EUR 119,944/QALY). Based on the information provided by Uhrmann’s [59] study in a German setting, the ICUR calculated for this ATMP was about EUR 156,853/QALY, with the threshold set at EUR 359,833/QALY. Hence, Uhrmann et al. considered voretigene neparvovec cost-effective versus SoC, despite its high cost [59]. There was a similar case in the United Kingdom. Hence, in this setting, for voretigene neparvovec, higher thresholds, ranging from EUR 124,607/QALY to EUR 373,821/QALY, are accepted as per the use of the ATMP for ultra-orphan conditions [60]. Zimmermann et al. [66] estimated that the incremental cost per QALY calculated for voretigene neparvovec (i.e., EUR 555,631/QALY from the US perspective) was not cost-effective even if considering higher thresholds related to ultra-rare diseases. Additionally, it was observed that only one study falls within the dominant quadrant [58], while as many as two studies [66,69] are positioned under the highest set threshold of EUR 150,000/QALY. Despite significantly higher costs, the highest incremental QALY observed was only around 10. Considering the three ICERs included in this study, the median ICER for voretigene neparvovec in the treatment of retinal diseases due to the RP E65 mutation is EUR 114,003 per QALY.

This ATMP is considered cost-effective only when a higher willingness-to-pay threshold is applied, consistent with ultra-rare disease valuation frameworks, since the condition it treats affects a very small patient population.

## 4. Discussion

This systematic review comprises the literature that evaluates the cost-effectiveness of ATMPs in rare disease treatment, specifically detecting the three most frequent macro-diagnoses: rare blood cancers (B-cell acute lymphoblastic leukemia-ALL, diffuse large B-cell lymphoma-DLBCL, multiple myeloma-MM, mantle cell lymphoma-MCL, follicular lymphoma-FL, and hemophilia B); spinal muscular atrophy type I- SMA; and retinal diseases with RP E65-mutation treatment. Main findings indicate that, while ATMPs approved for these rare diseases generally incur higher costs, they also provide superior QALYs compared to their standard of care or best supportive care, or alternative treatments.

More in-depth, tisagenlecleucel has emerged as a particularly promising option for rare blood disease treatment, such as ALL and DLBCL. Despite the availability of numerous therapeutic alternatives, this ATMP demonstrates significantly higher incremental QALYs, often ranging from 2 to 10 compared to the standard of care (SoC) [23,43,44,45,46,47,48,49,61,63,65,67,69,70,72,78,79]. These substantial health gains justify their elevated incremental costs in most healthcare settings, supporting their cost-effectiveness across diverse economic evaluations [80]. Moreover, these findings suggest that tisagenlecleucel, despite its high upfront costs, may result in long-term savings for healthcare systems by reducing the burden of disease and the need for subsequent interventions. Corroborating our findings, several studies have already evaluated the clinical efficacy of tisagenlecleucel, demonstrating its highly effective outcomes, in terms of short-term and long-term survival for the ALL treatment in both pediatric and adult populations [81,82,83,84,85]. Moreover, in addition to providing a therapeutic alternative that ensures safety and efficacy for patients enrolled in clinical trials, real-world evidence has shown that the results obtained with tisagenlecleucel are closely aligned with those observed in pivotal trials, further reinforcing its efficacy [86]. From a cost perspective, another systematic review by Andrade et al. has also assessed the cost-effectiveness of tisagenlecleucel for ALL treatments in children and young adults [80]. This review also highlighted that, although tisagenlecleucel is significantly more expensive than conventional therapies, it performed favorably on the ICER, remaining below the threshold of $100,000/QALY. This also confirmed that tisagenlecleucel is a more effective strategy than traditional small molecules and biological therapies in terms of life years and QALYs gained, making it a valuable option. A similar case is observed with axicabtagene ciloleucel, which also demonstrates strong cost-effectiveness profiles for the DLBCL treatment. However, its incremental QALY gains are lower than those observed with tisagenlecleucel for ALL and DLBCL. Importantly, therapy remains cost-effective from first- to third-line treatments, with its ICER estimates below the threshold set at EUR 100,000/QALY [20,23,30,32,35], but the cost-effectiveness tends to decline in later treatment lines. This could probably be explained due to reduced efficacy and smaller incremental health benefits in patients with more advanced DLBCL disease. Hence, a Chinese study showed that axicabtagene ciloleucel is not cost-effective as a second-line therapy in their setting (ICER: EUR 35,764/QALY; WTP: EUR 29,450/QALY), but this is not the case from the US perspective (ICER: EUR 111,318/QALY; WTP: EUR 117,318/QALY) [32].

Moreover, a recent systematic review by Thavorn et al. [87] showed similar results even if investigating only chimeric antigen receptor T-cell therapies for hematologic and solid malignancies. Hence, their cost-effectiveness plane underscores the same conclusions regarding the pharmacoeconomic profile of axicabtagene ciloleucel and tisagenlecleucel, emphasizing that studies evaluating the profile of axicabtagene ciloleucel are more concentrated until incremental QALY 3, while tisagenlecleucel achieves the highest concentration from 8 to 11 incremental QALY. Also, the CE plane clearly shows a positive economic profile of brexucabtagene autoleucel in MCL and ALL treatments, which is consistent with the findings of our study. Compared to the cost-effectiveness plane, our study supports a better cost-effectiveness profile of ciltacabtagene autoleucel in MM treatment, concluding that the new studies have been conducted and have been included in our research.

Regarding hemophilia B treatment with ATMPs, etranacogene dezaparvovec has been evaluated economically in only one study in Germany. This study classified it as cost-effective when compared to the current standard of care, coagulation factor IX (FIX) therapy [42]. The limited number of evaluations can be attributed to the novelty of the therapy and its recent market approval, a necessity of lifelong administration, as emphasized in hemophilia B treatment guidelines [88]. The recent study conducted by Pipe et al. [89] demonstrated a better safety and efficacy profile of etranacogene dezaparvovec compared to FIX therapy, improving the quality of life in patients with hemophilia B.

About the treatment of SMA type I with ATMPs, data from real-world settings exist to support the efficacy of the treatment, but it seems that its high cost may be a limitation in some settings [90]. Specifically, Dean et al., 2021 [77] identified it as a dominant option for SMA type I in patients under two years of age, with ICERs below US thresholds. However, in Australia, onasemnogene abeparvovec was not cost-effective (2 QALYs gained; high ICER), reflecting regional variations in healthcare economics [76]. This variability across healthcare systems underscores the need for tailored approaches to reimbursement policies.

Regarding retinal rare disease, a similar non-homogeneous situation was identified as per the SMA treatment with ATMPs. Hence, economic evaluations identified in this matter included voretigene neparvovec, a novel therapy approved for IRD patients with a rare biallelic RPE65 mutation, demonstrating an important distinction between high incremental costs and low incremental QALYs. Specifically, ICER analyses estimate incremental QALY gains of approximately 1.3 years compared to standard care [91]. While therapy offers clinical benefits, its pharmacoeconomic profile indicates that it is unlikely to represent good value for money relative to alternative treatments under traditional thresholds. This raises significant concerns regarding its economic feasibility in routine clinical practice.

The heterogeneity in the cost-effectiveness profiles of ATMPs for ultra-rare retinal diseases and SMA treatments, compared to the positive cost-effectiveness profiles of ATMPs for rare blood diseases, can be attributed to the small sample sizes related to these rare conditions. Also, a study by Khoa Ho stated that onasemnogene abeparvovec and voretigene neparvove had an additional challenge because they did not know the final price of this therapy, so they used assumed values to examine the cost-effectiveness of this therapy at different price levels [92]. NICE guidelines allow the discount rate for cost as well as an effectiveness of 1.5% for therapies that restore patients to near full or full health, acknowledging their transformative potential, as is the case with voretigene neparvovec and onasemnogene abeparvovec [7]. Due to their highly specific characteristics, these medicines are included in NICE’s ‘Highly Specialised Technologies’ program [93,94].

In comparison with the study by Thavorn et al., the highest incremental QALY observed for the treatment of ATMPs targeting hematological and solid malignancies is approximately seven. In contrast, Figure 4 shows that, in the evaluation of rare and ultra-rare diseases, the incremental QALY gain for onasemnogene abeparvovec reaches a high value of 22, an outlier that reflects long-term, exceptional benefit to the patient, despite the high cost. Among the 61 studies included in the analysis of treatments for rare diseases, 25 reported incremental QALY gains ranging from 5 to 22. The mean incremental QALY of the included studies is four, which is significantly higher compared to other drugs with an orphan designation [95], and is also higher than the incremental QALY in chronic conditions [96]. Some of the outliers identified in this cost-effectiveness analysis can be explained by a combination of methodological factor such as the following: (i) exceptionally high treatment costs, particularly for certain gene therapies such as voretigene neparvovec and onasemnogene abeparvovec, were associated with relatively modest QALY gains, resulting in unfavorable ICERs; (ii) also, low incremental health benefits were sometimes driven by conservative modeling assumptions, including the use of short time horizons or restrictive utility estimates; and methodological factors (iii) due to heterogeneity in the economic models such as differences in discount rates, WTP thresholds, or structural approaches (e.g., Markov models vs. partitioned survival models).

Given the uniqueness of this study, it is important to locate it in terms of the existing literature. Previous research by Olry de Labry-Lima et al. aimed to identify and provide a critical review of ATMP economic analysis [10]. On the other side, Thavorn et al. focused on the cost-effectiveness of chimeric antigen receptor T-cells for oncology patients’ treatment [87], while Khao Ho et al. conducted a systematic review that summarizes cost-effectiveness evidence of potential curative gene therapies and identified their challenges [92]. Furthermore, in 2020, Lloyd et al. published a systematic review on the cost-effectiveness evidence of all authorized ATMPs [7]. The strengths of our study lie in its comprehensive overview of the pharmacoeconomic profiles of ATMPs for treating rare and ultra-rare diseases, as well as in the sensitivity analysis conducted using three thresholds representing different healthcare system contexts, to support policymakers in making decisions.

This study has several limitations. This systematic review exclusively evaluates the economic profiles of EMA-approved ATMPs, and the included studies were sourced globally, to obtain a global framework across diverse economic settings and perspectives. Hence, most studies originated from high-income countries, reflecting the limited accessibility of these therapies in low- and middle-income countries. We recognize that limiting the study to open-access articles may lead to selection bias due to the potential exclusion of relevant papers. Gray literature, reports, and review articles were defined as exclusion criteria, which could cause an incomplete overview. To obtain complete and accurate conclusions on the assessment of the quality of evidence, due to potentially insufficient methodological information, the included study types are more suitable for evaluation using the mentioned checklist. This limitation was accounted for with the risk of bias assessment, where studies with these conditions received lower scores. Although the quality of evidence was assessed using a validated checklist and a double-blind methodology, some degree of subjectivity may have influenced the final estimations. One of the limitations lies in the significant number of studies with limited generalizability, thereby reducing the applicability of the results in general clinical practice. This could be explained by the nature of rare and ultra-rare diseases [19,24,39,49,50,70,76,77]. The quality of effectiveness evidence was low in some cases, but this was caused by the inherent limitations of clinical trials in the field of rare diseases, such as small sample sizes and single-arm study designs. Finally, when studies reported economic profiles of ATMPs over multiple time horizons, the longest horizon was selected to ensure the most robust results. Despite these efforts, uncertainties could remain, particularly regarding the outcomes and economic sustainability of these advanced therapies. It is necessary to consider social, health, and economic differences between countries, which are impossible to adjust and could potentially affect the results. Most of the included studies apply a health system perspective during evaluation, while in the case of orphan ATMP therapies, pricing decisions should include a broader societal perspective, considering the significant impact of these therapies not only on patients but also on their families, caregivers, and society [6]. These findings support the economic value of several ATMPs despite their high acquisition costs, particularly when long-term health gains are accounted for. In particular, the unique characteristics of ATMPs, such as their high upfront costs and often one-time administration, coupled with their potential for long-term, transformative benefits, necessitate robust and adaptable policy approaches. In fact, the findings of this study highlight that, despite the significant acquisition costs of ATMPs, their economic value is frequently supported by substantial gains in QALYs, particularly when considering long-term health impacts. This underscores the critical need for evidence-informed policy strategies to ensure timely and equitable access to these innovative therapies.

Given the innovative nature, high costs, and uncertainty in long-term effectiveness, policymakers must carefully balance innovation with system sustainability. To address these complexities, Managed Entry Agreements (MEAs) and Value-Based Pricing models should be prioritized. These approaches are crucial for mitigating financial uncertainty and aligning costs with actual clinical outcomes [97,98]. It is particularly pertinent to recognize that for ultra-rare diseases, the cost per patient tends to be even higher, and traditional WTP thresholds may be insufficient [99,100]. This often necessitates higher thresholds, or additionally dedicated programs such as those implemented by the British National Institute for Clinical Excellence (NICE) for Highly Specialised Technologies (HST) [93,94].

Furthermore, while most included studies adopt a healthcare system perspective, pricing decisions for orphan ATMP therapies should incorporate a broader societal perspective [101,102]. This involves considering the substantial impact these therapies have not only on patients but also on their families, caregivers, and society.

Therefore, there is an unmet need to develop more evidence to inform resource allocation and pricing negotiations. This approach will ensure that the real-world value of these high-cost therapies is considered. This ideal approach will ensure that the high-cost therapies will remain accessible to the patients who stand to benefit most from them [103]. Hence, in this scenario, adaptive reimbursement frameworks can evolve as more real-world evidence becomes available, potentially incorporating performance-based agreements that link payment to long-term patient outcomes, sharing financial risk between payers and manufacturers.

## 5. Conclusions

The systematic review presents a narrative synthesis of the overall economic profile of EMA-authorized ATMPs in rare disease treatment. The findings emphasize notable differences across therapeutic areas. Specifically, ATMPs for rare blood diseases, such as tisagenlecleucel and axicabtagene ciloleucel, generally showed positive cost-effectiveness profiles, with substantial health gains justifying their elevated costs in most healthcare settings. Similarly, etranacogene dezaparvovec represents a promising innovation for hemophilia B, offering a cost-effective alternative to lifelong coagulation factor IX therapy, though the evidence remains limited due to its recent approval. Ciltacabtagene autoleucel demonstrated a more favorable cost-effectiveness profile compared to idacabtagene vicleucel for the same indication. In contrast, the cost-effectiveness of ATMPs for spinal muscular atrophy (onasemnogene abeparvovec) and inherited retinal diseases (voretigene neparvovec) is less consistent, largely influenced by variations in healthcare contexts, WTP thresholds, and the unique characteristics of these rare conditions.

The findings underscore the need for a more distinct approach to assessment of cost-effectiveness, particularly in rare disease contexts where small sample sizes and limited evidence often make it difficult to assess long-term benefits, which could lead to unforeseen costs for policymakers. Policymakers must carefully weigh the costs and benefits of these therapies, considering both the economic constraints of healthcare systems and the real-world impact on patient outcomes.

Moreover, there is a pressing need for continued research to refine economic models, address data gaps, and develop strategies for reducing the costs of these therapies, ensuring that they are accessible to the patients who would benefit the most.

## Figures and Tables

**Figure 1 healthcare-13-01894-f001:**
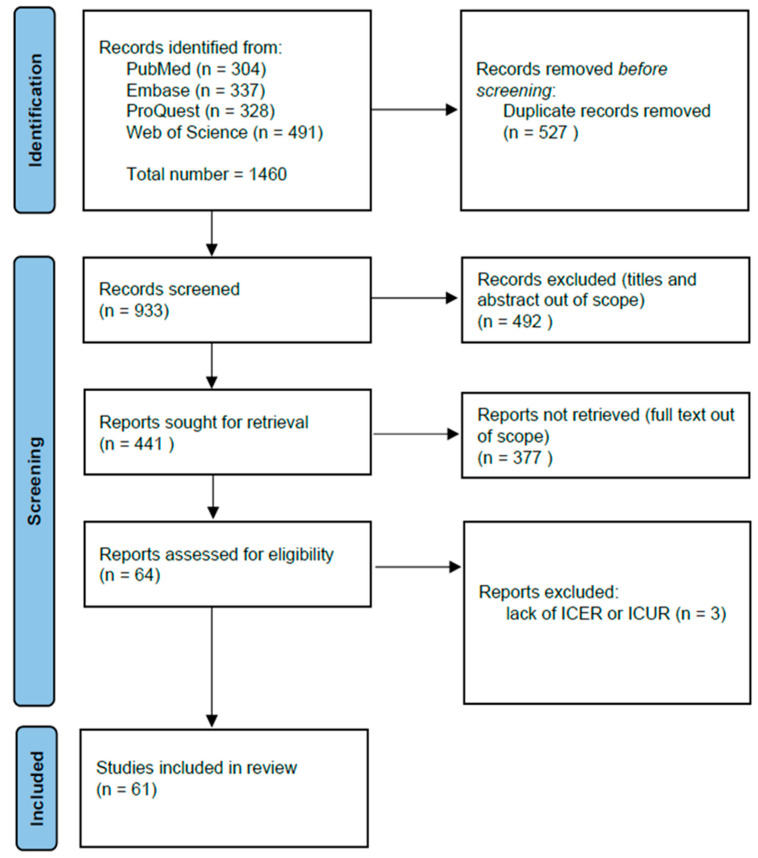
The PRISMA flow diagram of study selection.

**Figure 2 healthcare-13-01894-f002:**
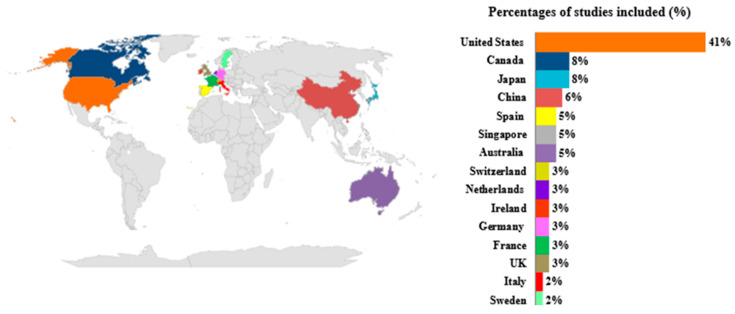
Global distribution of economic evaluations included in the review.

**Figure 3 healthcare-13-01894-f003:**
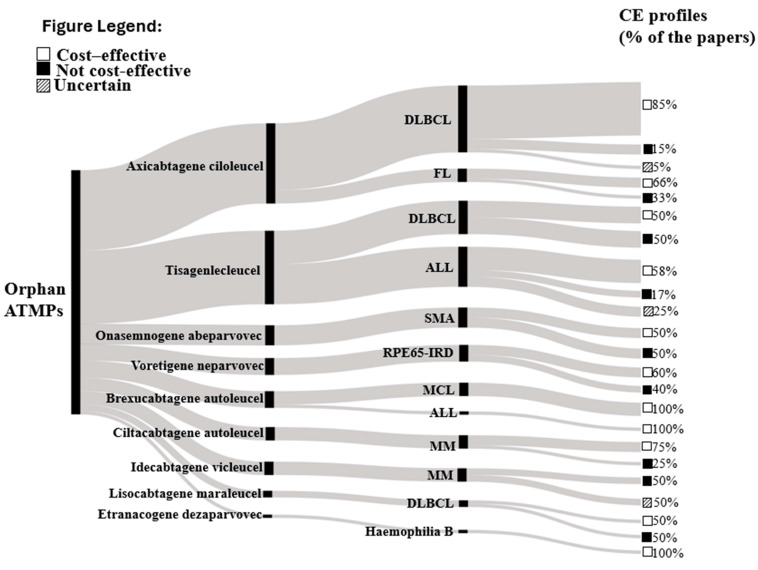
Sankey chart of the CE profile of ATMPs and their specific diagnosis. Footnotes: The Sankey diagram depicts the distribution of cost-effectiveness analyses of ATMPs. Each segment represents a particular ATMP, with the flow’s size indicating the number of analyses deemed cost-effective or not. It also shows the relative frequency of the ATMPs for each therapeutic indication. Abbreviations: DLBCL—Diffuse large B-cell lymphoma; ALL—Acute lymphoblastic leukaemia; FL—Follicular lymphoma; SMA—Spinal muscular atrophy; MCL—Mantle cell lymphoma; MM—Multiple myeloma; Cost-effectiveness (CE); RPE 65-IRD: Inherited retinal diseases due to RPE65 variants.

**Figure 4 healthcare-13-01894-f004:**
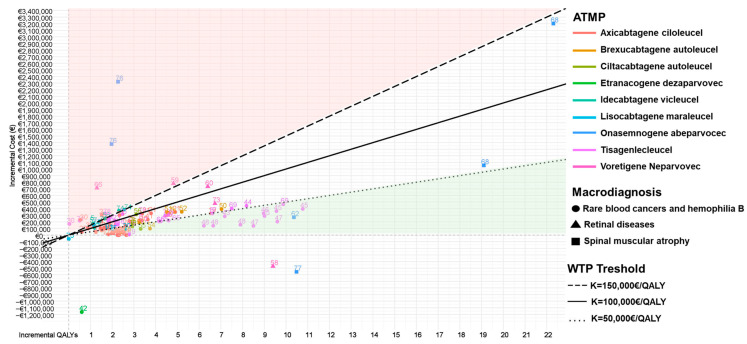
Cost-effectiveness plane stratified by ATMPs and macro-diagnosis. Notes: northeast quadrant: more effective but more costly; southeast quadrant: dominant = more effective and less costly; northwest quadrant: dominated = less effective and more costly; southwest quadrant: less effective but less costly.

**Table 1 healthcare-13-01894-t001:** Eligibility criteria.

Study Characteristics	Inclusion Criteria	Exclusion Criteria
Study type	Research articles, articles in the press, evaluation studies, evidence-based studies, comparative studies, economic evaluations, assessments, communications, and original investigations	Books, book reviews, commentaries, conference proceedings, rationale and/or design, corrections/retraction, dissertation/thesis, editorials, literature reviews, evidence reviews, scoping reviews, letters, periodicals, conference abstracts, study protocols, consensus, notes, systematic reviews, conference papers, erratum, short surveys, conference reviews, chapters, reports, case reports, case studies, review articles, clinical trials, viewpoint articles, rapid communication, gray literature
Publication stage	Published articles	Preprints; under publication articles
Publication availability	Full open-access articles	Non-open access articles; partial open access articles
Language	English	All other languages
Setting	All	Not applied
Time frame	All	Not applied
Patient (P)	Patients diagnosed with a rare disease eligible for treatment with ATMPs	Patients without a rare disease, or not eligible for ATMP treatment
Intervention (I)	Subjects treated with any ATMPs authorized by the EMA with an orphan designation. (withdrawn or not-renowned marketing authorizations ATMPs were not considered eligible) (Appendix A)	Patients treated with withdrawn, non-renewed, or non-orphan-designated ATMPs
Comparator (C)	Standard of care (SoC), best supportive care (BSC), and other therapies	Studies with inappropriate or unreported comparators
Outcomes (O)	Cost-effectiveness profile, cost and quality-adjusted life years gained, incremental cost-effectiveness ratio (ICER), and incremental cost-utility ratio (ICUR)	Studies not reporting relevant economic outcomes
Study design (S)	Economic evaluation, cost-effectiveness analysis, and cost-utility analysis	Non-economic study types

**Table 2 healthcare-13-01894-t002:** Study Overview and Key Characteristics.

International Nonproprietary Name	Therapeutic Indications *	ICER Mean	Incremental QALY Mean (SD)	References
Axicabtagene ciloleucel	Diffuse large B-cell lymphoma (DLBCL) (87%) Folicular lymphoma (FL) (13%)	EUR 59,183/QALY	2.43 (1.33)	[19,21,22,23,24,25,26,27,28,29,30,31,32,33,34,35,36,38,39,40,41]
Tisagenlecleucel	Acute lymphoblastic leukaemia (ALL) (60%) Diffuse large B-cell lymphoma (DLBCL) (40%)	EUR 55,013/QALY	5.64 (2.9)	[23,30,43,44,45,46,47,48,49,61,63,64,65,67,69,70,71,72,78,79]
Onasemnogene abeparvovec	Spinal muscular atrophy (SMA)	EUR 89,566/QALY	13.23 (9.50)	[62,68,77]
Voretigene neparvovec	Biallelic RPE65-mediated inherited retinal disease	EUR 115,572/QALY	5.98 (1.02)	[59,60,73]
Brexucabtagene autoleucel	Mantle cell lymphoma (MCL) (80%) Acute lymphoblastic leukemia (ALL) (20%)	EUR 53,774/QALY	4.40 (1.48)	[50,51,52,53,54]
Ciltacabtagene autoleucel	Multiple myeloma (MM)	EUR 58,511/QALY	3.07 (0.18)	[55,56,57]
Idecabtagene vicleucel	Multiple myeloma (MM)	EUR 138,877/QALY	1.59 (0.58)	[55,56,57,74]
Lisocabtagene maraleucel	Diffuse large B-cell lymphoma (DLBCL)	EUR 48,881/QALY	1.88 (0.21)	[30,75]
Etranacogene dezaparvovec	Hemophilia B	EUR −1,937,936/QALY	0.60	[42]

* The percentage of study distribution was calculated in the case of two or more ATMPs’ therapeutic rare indications. Notes: Due to significant variations in the mean values due to the outlier values, these latter have been excluded from the overview table to provide a clearer insight. However, for the drug Etranacogene dezaparvovec, no outlier exclusion was applied, as only one study for this drug was included. Complete information of each study is detailed in Appendix A.

## Data Availability

The original contributions presented in this study are included in the article/Appendix A. Further inquiries can be directed to the corresponding author.

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
