# Peer review of "Pharmacoeconomic Profiles of Advanced Therapy Medicinal Products in Rare Diseases: A Systematic Review"

_healthcare, 2025, doi:10.3390/healthcare13151894_

Round 1
Reviewer 1 Report
Comments and Suggestions for Authors
This manuscript has systematic reviewed the pharmacoeconomic profile of advanced therapies for rare disease. The review has made literature search and included 61 studies finally. Quality assessment was done. And the authors provide the full landscape of the pharmacoeconomic profiles of current available evidence. The topic is important is interesting. The method is detailed and reliable. The manuscript is well structured. The references are relevant. Although evidence synthesis is not formed, the review is important for the understanding the pharmacoeconomic profile of recent novel therapies and would be helpful for further therapy development.
I only have several minor comments.
1) According to PRISMA guideline, it should be specified in the title that this is "a systematic review"
2) Table 2. The structure should be improved. I suggest the authors to add a column about the items of inclusion and exclusion.
3) I can not find the citations of Table S1 and S2 in the maintext.
4) Figure 2. A bar column would be better.
5) The conclusion is abs should be more comprehensive. The detailed results should be in the result section.
Author Response
Dear Reviewer, on behalf of all co-authors we thank you for the valuable comments which we implemented in this new revised version of the manuscript. Please find detailed responses below and the corresponding revisions/corrections in track changes in the resubmitted files (see pdf file attached).

Reviewer 2 Report
Comments and Suggestions for Authors
This systematic review aims to synthesize the economic evaluation literature on European Medicines Agency (EMA)-authorized Advanced Therapy Medicinal Products (ATMPs) for rare diseases. The authors searched PubMed, Embase, Web of Science and ProQuest following PRISMA guidelines, included 61 studies, and extracted incremental cost-effectiveness ratios (ICERs/ICURs), standardizing all costs to 2023 € via the CCEMG-EPPI tool. The review reports that ATMPs for blood cancers (e.g. tisagenlecleucel, axicabtagene ciloleucel) generally show favorable cost-effectiveness profiles, whereas treatments for ultra-rare conditions (e.g. retinal diseases, SMA) yield mixed results. The study highlights differences in cost-effectiveness across indications and emphasizes policy considerations. Overall, the review addresses an important topic, but several methodological and reporting issues need clarification or improvement before acceptance.
Comments
-
Search Strategy and Inclusion Criteria (Potential Selection Bias): The eligibility criteria (Table 1) restrict inclusion to “Full open-access articles”. This restriction is highly unusual for a systematic review and risks excluding a substantial body of relevant literature (e.g. subscription-based journals, HTA reports). Limiting to open-access likely introduces selection bias and reduces comprehensiveness. I strongly recommend removing this restriction and instead obtaining full texts by interlibrary loan or contacting authors. The authors should clarify this choice and, if not changed, at least acknowledge it as a limitation. Otherwise, critical studies may have been missed. In addition, while the search date (October 4, 2024) is recent, the authors should ensure no important studies were published just after the search. If so, consider a brief update or mention of recent evidence.
-
Definition of “Cost-Effective” and Use of Thresholds: The manuscript states that profiles “with an ICER/ICUR exceeding 50% of the willingness-to-pay (WTP) threshold were considered cost-effective”. This phrasing is unclear and appears to contradict standard practice (normally an intervention is cost-effective if ICER is below the threshold). Please clarify this criterion. If the intent was to define cost-effective as ICER ≤ 0.5 × WTP, this unusual rule needs justification. Moreover, the authors then conduct sensitivity analyses at fixed thresholds (€50k, €100k, €150k) based on NICE’s Highly Specialized Technologies guidance. The rationale for these specific values (especially €150k/QALY) should be further explained, since different countries and diseases use different thresholds. It would strengthen the paper to discuss country-specific thresholds (e.g. 1–3× GDP per capita) or reference case thresholds. At minimum, acknowledge that applying the same three global thresholds to all studies ignores local decision rules and may affect the interpretation of “cost-effective”.
-
Data Conversion and ICER Calculation: Standardizing costs to 2023 € using the CCEMG-EPPI Inflation and PPP tool is a strength. To improve transparency, please detail how original currencies and years were handled. For instance, specify whether all studies reported base-year costs and currency; if not, how base years were determined. It would also be helpful to report a summary (perhaps in a supplement) of original costs and converted costs for key studies. Regarding ICERs: the authors calculated missing ICERs “as feasible based on the data provided”. Please briefly describe the formula or method (e.g. ΔCost/ΔQALY) and how data gaps (e.g. missing QALYs) were handled. Clarifying these steps is important for reproducibility. Finally, how were “dominant” results (cost-saving, negative ICER) treated in the narrative and figures? For example, one study found €–53,073/QALY (dominant), but the decision rule for dominance should be stated (e.g. interventions with negative ICERs were classified as cost-effective).
-
Quality Assessment and Risk of Bias: The authors used the Olry de Labry-Lima/ISPOR checklist to assess study quality and risk of bias, rather than the more common CHEERS or Drummond checklists. It would help readers to justify this choice. How does this tool compare to the CHEERS guideline, and why is it preferable here? The description of scoring (threshold of 75% for “higher quality”) and rating (moderate/low to high confidence) in the Supplement suggests a complex process, but this is not clearly explained in the main text. The manuscript should explicitly state who performed the quality assessment (e.g. two independent reviewers) and how discrepancies were resolved (it is briefly mentioned that “differences were discussed”). Consider summarizing the main domains (eligibility criteria, perspective, time horizon, etc.) and results (how many studies met each item) in a table or figure. The authors found that 55% of studies had “possibility for conflict of interest”, but it is unclear how COI was evaluated. Did they use funding or author affiliations? Clarify how COIs were identified (e.g. whether industry-sponsored). Given the prevalence of COI, the discussion should comment on whether studies with declared industry involvement showed different results. For instance, a sensitivity analysis or narrative comparison excluding industry-funded studies could be informative to assess bias.
-
PRISMA Reporting and Supplementary Material: The authors indicate adherence to PRISMA guidelines and provide a flow diagram (Figure 1). However, ensure all PRISMA 2020 items are addressed. For example, the search strategy for each database should be fully reported (the text mentions Table S3–S4 as containing search syntax). The number of records excluded at each stage is given in Figure 1, but PRISMA also recommends a list of excluded full-text articles with reasons. The text notes 3 full texts were excluded for lacking ICER data; it would improve transparency to list those references (perhaps in Supplement). Also, confirm that a PRISMA checklist is included in the supplementary materials. If not, adding it would strengthen compliance. Finally, check that all cited supplementary tables (S1–S7) are complete and properly referenced; for example, definitions (Table S1), search strategy (Tables S3–S4), and quality assessment (Tables S6–S7) are mentioned throughout but should be easily accessible.
-
Figures and Tables – Clarity and Consistency: The PRISMA flow (Fig.1), Sankey chart (Fig.3), and cost-effectiveness plane (Fig.4) are informative. Ensure that all figures and tables are clearly labeled. For Figure 4 (cost-effectiveness plane), the caption should explicitly label the axes (“Incremental QALYs” on x-axis, “Incremental Cost (€)” on y-axis) and define all quadrants (e.g. “dominant = lower cost, higher QALYs”) so that readers can interpret the scatter without confusion. The Sankey diagram (Fig.3) uses colors (green/red/yellow); please verify color-blind–friendly hues or include patterns if possible, and ensure the legend text is legible at publication size. In the text (e.g. [10†L820-L829]), some values are reported with different thresholds (e.g. 50% of studies for axi-cel in DLBCL, but 85% in the Results paragraph). Double-check that all percentages match the counts and that each figure legend matches the narrative.
Table 3 (characteristics of included studies) spans many pages. Confirm that column headings are unambiguous (e.g. specify the year of costs in the “Adjustment” column). In Table 4 (not fully shown here), ensure each ATMP’s ICER range is presented clearly. The current formatting in the review file (lines [7†L1301-L1309], [14†L1088-L1096]) seems disjointed; format issues may arise in peer-review PDF. For the final version, ensure consistent use of currency symbols and decimals (e.g. “€ 29,449/QALY” vs “€29,449/QALY”).
-
Analysis and Synthesis of Results: The decision not to meta-analyze due to heterogeneity is appropriate, but consider summarizing findings more quantitatively where possible. For example, instead of narrative statements (“85% of studies showed axi-cel cost-effective in DLBCL”), it might be useful to report median ICERs or interquartile ranges across studies for key ATMPs. Alternatively, a small table summarizing ICER ranges by indication (with number of studies below/above threshold) could help readers. The cost-effectiveness plane (Fig.4) does show clusters, but text descriptions (e.g. “axicabtagene ciloleucel has notably lower incremental costs”) could be supported by summary statistics. If space permits, the authors might report how many studies found dominance (cost-saving) vs dominated (worse outcomes) and how uncertainty was addressed in each study (were probabilistic sensitivity analyses common?).
-
Interpretation and Generalizability: The discussion and conclusions highlight that blood cancer ATMPs tend to be cost-effective while others vary. This is a key contribution. However, the authors should temper conclusions by noting limitations. For instance, most studies were from the US; thus applicability to European decision-making may be limited. The influence of different healthcare systems, price negotiations, and comparator practices should be acknowledged. Likewise, the high heterogeneity in methods (as noted) implies caution in comparing ICERs directly. The conclusion suggests policy strategies to balance innovation and affordability, but could be strengthened by explicitly tying recommendations to findings (e.g. using managed entry agreements or value-based pricing for ATMPs). Finally, it would be helpful to mention any potential publication bias or “funding bias,” given the high rate of possible industry involvement, even though formal assessment of bias (e.g. funnel plot) is not applicable.
Overall, this manuscript addresses a timely topic with a systematic approach. Addressing the above major points—especially the open-access limitation, threshold definitions, and clarity in methods—will strengthen its rigor and credibility. The minor edits will improve readability and adherence to journal style. With these revisions, the paper would make a valuable contribution to the pharmacoeconomics literature.
Author Response
Dear Reviewer, thanks for reviewing the manuscript. We entirely revised the whole manuscript and resubmitted the implemented version. Please, find detailed responses below and the corresponding revisions/corrections in track changes in the resubmitted files (see pdf file attached).

Reviewer 3 Report
Comments and Suggestions for Authors
This manuscript provides a comprehensive systematic review of the cost-effectiveness of EMA-approved ATMPs for rare diseases. The methodology is generally sound and well-documented, the results are detailed and well-structured and the discussion and conclusion effectively interpret the findings. Overall, this is a valuable contribution to the field and supports informed decision-making around the economic evaluation of ATMPs.
- Limiting to only open-access articles could introduce publication bias and exclude high-quality studies. This criterion is not commonly justified in systematic reviews and should be explained or reconsidered.
- Exclusion of gray literature might limit comprehensiveness. This is especially relevant in pharmacoeconomics, where such sources are common.
- While the search strategy is said to be in the supplementary material, it would be helpful to briefly summarize key terms in the main text.
- “Studies scoring above an arbitrary threshold of 75% were considered to have higher reporting quality.” How was the 75% threshold derived?
- The discussion mostly restates the results in a descriptive way and should be modified.
Author Response

(The authors gave the same response as above.)

Reviewer 4 Report
Comments and Suggestions for Authors
This systematic review is a timely and valuable contribution to the growing field of health economics and personalized medicine.
Abstract & Introduction
Several grammatical and stylistic issues reduce clarity e.g., “cost-effectiveness profile” is repeated excessively. Replace “demonstrated strong cost-effectiveness profiles” with “were found to be cost-effective in a majority of studies.”
Some sentences are too long or awkwardly constructed. Examples:
-
“However, the question arises as to whether the price justifies the achieved outcome.” → Simplify to: “However, it remains unclear whether these prices are justified by clinical outcomes.”
-
“This pharmaceutical group has led to elevated costs because of its innovative nature...” → Reword: “ATMPs are associated with elevated costs, largely due to their innovative, often one-time administration.”
Clarify if the QALY gains reported (e.g., “1.5 to 10”) are per patient and over what time horizon.
Materials and Methods
Several minor grammatical errors, awkward phrasing, and sentence structure problems occur (e.g., “made. Studies were scored…” in line 132).
Some long sentences (e.g., lines 110–139) would benefit from breaking into shorter, clearer statements.
The checklist adapted from Olry de Labry-Lima et al. is useful but not widely recognized or validated like CHEERS 2022. Either justify the choice more clearly or include a dual assessment using CHEERS as well.
Data extraction was done in Excel, which is acceptable, but use of standardized systematic review software (e.g., Covidence, Rayyan) could enhance transparency and reproducibility.
Although risk of bias is assessed, publication bias is not addressed, which is relevant when excluding gray literature. Briefly discuss this limitation in the methods or results.
While multiple reviewers are used, no kappa statistic or measure of inter-rater reliability is mentioned. Include a kappa or percentage agreement value to enhance credibility.
Discussion & Conclusions
Many sentences are long, dense, and awkwardly phrased. Example: “This ATMP has been evaluated economically in only one study, which classified it as cost-effective in Germany…”
While detailed, the section would benefit from summary insights after each therapeutic category (e.g., blood cancers, SMA, retinal diseases) to consolidate key findings.
The paragraph about Figure 4 and incremental QALYs up to 22 is interesting but vague. It’s unclear:
-
-
What specific interventions achieved 22 QALYs?
-
Are these outliers, or representative?
-
Several statements about high upfront costs and long-term benefits are repeated verbatim or nearly so across multiple paragraphs. This dilutes impact.
The final paragraph before the conclusion attempts to compare the study to existing literature but doesn’t offer a strong synthesis or implication for future research or policy. Consider ending with a statement that re-emphasizes novelty, potential impact, and recommendations for decision-makers.
Results
There is a numerical inconsistency in line 179: 492 abstracts are screened after removing 527 duplicates, yet this implies a total of 1,019 papers, not 1,460. Clarify this discrepancy.
“n = 337 papers without full-text availability were removed” suggests a large proportion of missing full-texts—was this due to paywalls or unavailability? This needs justification or discussion.
Risks of bias are listed (assumption bias, study design bias, extrapolation bias) but not exemplified or quantified.
The “55% possibility for a conflict of interest” is a strong claim but not contextualized (e.g., is this due to industry funding?).
The discussion of discount rates is brief and general - no specific examples beyond “up to 5%” are offered. Which countries used which rates?
Some studies are described as “outliers” without sufficient explanation of why they deviate - what factors (price, indication, model choice) explain this.
The phrase “leading to completely different conclusions in the same settings” (lines 293–294) is vague. Why were the results so different? This deserves deeper analysis, perhaps in the Discussion.
Author Response
Dear Reviewer, thanks for reviewing the manuscript. We entirely revised the whole manuscript and resubmitted the implemented version. Please, find detailed responses below and the corresponding revisions/corrections in track changes in the resubmitted files. (Please check the attached PDF file.)

Round 2
Reviewer 3 Report
Comments and Suggestions for Authors
All of previous comments have been fully addressed in the revised manuscript, and it is now in good shape for publication.
Author Response
All of previous comments have been fully addressed in the revised manuscript, and it is now in good shape for publication.
Response: Dear reviewer, many thanks for your consideration of our research study.
Reviewer 4 Report
Comments and Suggestions for Authors
The authors have revised thoroughly their manuscript according to my previous suggestions. I only have minor comments:
The manuscript includes long, raw lists of ICERs in the results. It would be better to condense the most critical ICER data in a summary table by ATMP and indication (possibly in the main text), leaving full details in the supplement. Color-code thresholds (e.g., red > €150,000/QALY, green < €50,000/QALY) for quick visual impact.
The discussion primarily reiterates results with limited exploration of implications. The authors could expand the discussion on policy implications, such as implications for reimbursement frameworks or value-based pricing.
Author Response
Dear reviewer, thank you very much for your feedback. Please find detailed responses below and the corresponding revisions/corrections in the resubmitted files.
Comment 1: The manuscript includes long, raw lists of ICERs in the results. It would be better to condense the most critical ICER data in a summary table by ATMP and indication (possibly in the main text), leaving full details in the supplement.
Response 1: We have created a new Table 2 that summarises the average incremental cost values with the pertaining incremental QALYs for each of the ATMPs anlysed, considering the diagnosis of rare disease. The old, detailed Table 2 was transferred into the supplementary material. We have clearly made all these relevant changes in the manuscript regarding the citation of tabs and figures.
Comment 2: Color-code thresholds (e.g., red > €150,000/QALY, green < €50,000/QALY) for quick visual impact.
Response 2: As suggested, in the cost effectiveness plan (Figure X) we coloured the plan red above the threshold of €150,000/QALY and green below the threshold of €50,000/QALY for better clarity.
Comment 3: The discussion primarily reiterates results with limited exploration of implications. The authors could expand the discussion on policy implications, such as implications for reimbursement frameworks or value-based pricing. "
Response 3: We agree with your suggestions regarding the implementation of the discussion section with more considerations regarding implications for reimbursement frameworks and value-based pricing. Hence, we’ve now proceeded to implement the discussion with these aspects in lines 555 to 587. Specifically, as follows:
“(..) These findings support the economic value of several ATMPs despite their high acquisition costs, particularly when long-term health gains are accounted for. In particular, the unique characteristics of ATMPs, such as their high upfront costs and often one-time administration, coupled with their potential for long-term, transformative benefits, necessitate robust and adaptable policy approaches. In fact, the findings of this study highlight that, despite the significant acquisition costs of ATMPs, their economic value is frequently supported by substantial gains in QALYs, particularly when considering long-term health impacts. This underscores the critical need for evidence-informed policy strategies to ensure timely and equitable access to these innovative therapies.
Given the innovative nature, high costs, and uncertainty in long-term effectiveness, policymakers must carefully balance innovation with system sustainability. To address these complexities, Managed Entry Agreements (MEAs) and Value-Based Pricing models should be prioritized. These approaches are crucial for mitigating financial uncertainty and aligning costs with actual clinical outcomes. It is particularly pertinent to recognize that for ultra-rare diseases, the cost per patient tends to be even higher, and traditional WTP thresholds may be insufficient. This often necessitates higher thresholds, or additionally dedicated programs such as those implemented by the British National Institute for Clinical Excellence (NICE) for Highly Specialised Technologies (HST).
Furthermore, while most included studies adopt a healthcare system perspective, pricing decisions for orphan ATMP therapies should incorporate a broader societal perspective. This involves considering the substantial impact these therapies have not only on patients but also on their families, caregivers, and society.
Therefore, there is an unmet need to develop more evidence to inform resource allocation and pricing negotiations. This approach will ensure that the real-world value of these high-cost therapies is considered. This ideal approach will ensure that the high-cost therapies will remain accessible to the patients who stand to benefit most from them. Hence, in this scenario, adaptive reimbursement frameworks can evolve as more real-world evidence becomes available, potentially incorporating performance-based agreements that link payment to long-term patient outcomes, sharing financial risk between payers and manufacturers”.